# Elbow Arthroscopy for the Treatment of Radial Head Fractures: Surgical Technique and 10 Years of Follow Up Results Compared to Open Surgery

**DOI:** 10.3390/jcm12041558

**Published:** 2023-02-16

**Authors:** Enrico Guerra, Alessandro Marinelli, Fabio Tortorella, Michelle Dos Santos Flöter, Alice Ritali, Andrea Sessa, Giuseppe Carbone, Marco Cavallo

**Affiliations:** 1Shoulder and Elbow Unit, IRCCS Istituto Ortopedico Rizzoli, 40136 Bologna, Italy; 2Orthopedics and Traumatology Department, Kantonsspital St. Gallen, 9000 St. Gallen, Switzerland

**Keywords:** elbow arthroscopy, radial head fracture, portals, radial head ARIF, elbow fractures

## Abstract

Background: This study aimed to describe the ARIF (Arthroscopic Reduction Internal Fixation) technique for radial head fractures and to compare the results with ORIF (Open Reduction Internal Fixation) at mean 10 years. Methods: A total of 32 patients affected by Mason II or III fractures of the radial head who underwent ARIF or ORIF by screws fixation were retrospectively selected and evaluated. A total of 13 patients were treated (40.6%) by ARIF and 19 patients (59.4%) by ORIF. Mean follow-up was 10 years (7–15 years). All patients underwent MEPI and BMRS scores at follow-up, and statistical analysis was performed. Results: No statistical significance was reported in Surgical Time (*p* = 0.805) or BMRS (*p* = 0.181) values. Significative improvement was recorded in MEPI score (*p* = 0.036), and between ARIF (98.07, SD ± 4.34) and ORIF (91.57, SD ± 11.67). The ARIF group showed lower incidence of postoperative complications, especially regarding stiffness (15.4% with ORIF at 21.1%). Conclusions: The radial head ARIF surgical technique represents a reproducible and safe procedure. A long learning curve is required, but with proper experience, it represents a tool that might be beneficial for patients, as it allows a radial head fracture to be treated with minimal tissue damage, evaluation and treatment of the concomitant lesions, and with no limitation of the positioning of screws.

## 1. Introduction

Radial head fractures are a common occurrence, with a reported incidence of 2.5 per 10,000 per year [1]. These fractures are usually caused by a fall onto outstretched hand [2], and about 30% of cases include associated injuries. The radial head is important because it functions as the elbow’s secondary stabilizer for valgus stresses as well as a primary stabilizer for axial loads, with the interosseus membrane as a secondary stabilizer [3]. Its shape has highly variable angles, dimensions and curvature [4], so even though improvements have been achieved in anatomical prosthesis design, preserving the native radial head when possible remains the gold standard. For the same reasons, it is important to avoid radial head resection as much as possible during surgeries [5].

The most common classification of radial head fractures was introduced by Mason, was then modified by Hotchkiss [6], and then further modified by Bromberg and Morrey [7]. Radial head fractures type 1, with a displacement inferior to 2 mm, are usually treated conservatively if there is not a mechanical block during pronation–supination movements. Controversies arise for type II, however, when a displacement between >2 mm and <5 mm occurs without mechanical block [8,9,10], because conservative treatment can be indicated even if a radial head deformity is present. Type II and III are classically treated with ORIF (Open Reduction Internal Fixation), partial resection of the radial head (when less than 30% of the articular surface is involved) or radial head prosthetic replacement [11,12].

Over recent years, advances in elbow arthroscopy instruments, surgeons’ skills, and increased knowledge of local anatomy have all led to an increment improvement of radial head fractures fixation, [13] allowing fractures that would have been treated by ORIF [14] to be treated by ARIF.

Studies have shown that it can be technically challenge to arthroscopically expose the radial head to allow for appropriate screw placement. To obtain stable screw fixation, it is important not only to reach the desired radial head portion, but also to achieve an optimal Kirschner wires entry point and direction and to precisely direct cannulated screws [15,16]. Recent studies have shown how, with skilled use of a combination of anteromedial, anterolateral, and midlateral portals, it is possible to reach a 360° exposure of the radial head for arthroscopic fracture fixation [17]. Furthermore, the distalization of anteromedial and anterolateral portals is useful to obtain a biomechanical stronger fixation [17]. 

To our knowledge there are no current studies that compare radial head ARIF and ORIF. Therefore, the aim of this study was to describe the technique used to perform radial head ARIF and to compare the results after 10 years by analyzing patients who received ARIF to those who received ORIF.

We hypothesized that skilled surgeons who were trained on elbow arthroscopy would be able to perform ARIF of the radial head with a comparable surgical time, better early functional outcome, and a lower rate of postoperative complications than for ORIF.

## 2. Materials and Methods

A retrospective study was conducted on patients who underwent surgery between June 2007 and October 2015. All the procedures were performed by the same skilled surgeon (EG).

Patients who were eligible for the study were those who underwent arthroscopic or open field radial head fixation with screws and who did not have ligament injuries or coronoid associated fractures. Indications for surgery were: mechanical block in pronosupination movements; two-part fractures with displacement greater than 5 mm if involving head or greater than 4 mm if involving the neck; and fractures that had multiple fragments but were still treated with screw fixation. Patients treated with plate and screws fixation or radial head removal or replacement were excluded. 

Patients’ demographics are reported in Table 1. 

### 2.1. Surgical Planning

Every patient underwent a pre-op CT scan with multiplanar 2D and 3D reconstructions to study the number, size, and dislocation of fracture fragments, as well as bone impaction and involvement of the ‘safe zone’.

ARIF was performed in 13 cases where the pre-operative plan was to perform a screw fixation, and there was an absence of gross associated injuries or Fracture type Mason II and III without LCL at varus stress test under anesthesia. A total of 19 patients who underwent ARIF were compared to a group of patients who underwent ORIF with the same indications.

### 2.2. ARIF Surgical Technique Description

Elbow varus–valgus stability was tested with the patient supine under anesthesia (brachial plexus block or general anesthesia). 

Patients were positioned in lateral decubitus with the affected arm supported by a dedicated holder, and a tourniquet was inflated at 250 mmHg. Joint landmarks were drawn on the skin. An articular lavage was performed by a spinal needle through the ‘soft spot’ in the posterolateral portion of the elbow.

Three standard anterior portals were routinely performed: anteromedial, anterolateral and proximal anterolateral. The pump maintained an intra-articular pressure in an interval between 30 and 40 mmHg.

#### 2.2.1. 1st Step: Fracture Visualization and Reduction

The arthroscopic camera was inserted in the anteromedial portal with the shaver in the anterolateral; at the same time, an elevator kept the joint capsule open from the anterolateral proximal portal.

The hematoma was removed, and the fracture was visualized from both the anteromedial and anterolateral portal to confirm CT scans.

The fracture was then mobilized and reduced using two probes that were introduced from the anterolateral and proximal anterolateral portals along with pronation and supination movements. If necessary, a third probe could be inserted through the midlateral portal/soft spot to achieve and/or maintain the reduction. Some fractures can be difficult to reduce because the radial head fragment can be dislocated either through the anterior capsule or in the posterior aspect of the elbow. As no fractures with high dislocations were present in the cases in this series, no cases required the capsule to be opened, and none were converted into open field surgeries. This could occur because associated ligamentous or coronoid lesions are often present when the fragment is dislocated through the capsule or in the posterior aspect of the elbow, and these associated injuries were an exclusion criteria for the study.

Once reduction was achieved, the best position between pronation and supination to perform definitive fragment fixation was chosen, and the fragments were stabilized with temporary 1.4 mm K-wires. 

#### 2.2.2. 2nd Step: Working Position and Fixation

Using specimen studies, the elbow and wrist ESSKA committee showed that the entire radial head circumference is approachable for fixation, and that modified anteroinferior portals are more effective for performing a correct radial head fracture ARIF1 [17,18]. 

Fractures can be divided according to the position of the fragment in relation to the “safe zone” (i.e., the part of the radial head that does not have contact with the small sigmoid notch). Ideally, the radial head is divided into two halves in the neutral position: the lateral half (which includes the safe zone) and the medial half.

Fractures of the lateral half can be fixed from the anterolateral portal (*working position AL*) by placing the forearm in pronation and examining from the AM portal (Figure 1).

The retractor and the probe help to maintain the reduction and keep the workspace open. It may be useful to temporarily fix the fracture by using a K-wire either posteriorly (soft spot portal) or anterolaterally (anterolateral portal).

An accessory lateral portal from a more distal location could be useful for placing the screws parallel to the radial head surface. A small 5 mm cannula was needed to protect the soft tissues from the rotating tools (K-wire, drill, screwdriver) and to prevent the thin K-wire from bending or breaking. The K-wire was inserted through the cannula, and then the fixation procedure was performed (measurement, drilling, screw).

Fractures of the lateral half could be alternatively fixed by working posteriorly (working position PL) as described by Rolla et al. [16] (Figure 2). 

Posterolateral and midlateral portals were created. From the superior view, the shaver and probe created the necessary space to reduce the fragment. The fixation technique was then completed by following the steps previously described. 

For medial half fractures, it was helpful to change the view (working position AM, Figure 3).

While viewing from the anterolateral portal, the cannula was inserted through the anteromedial portal. The procedure for the fixation was the same, and no other accessory portals were needed. 

Following these considerations, the three working positions can be summarized as in Table 2.

All fractures were fixed using headless cannulated screws (Micro Acutrak screws, Acumed) with lengths varying from 14 mm to 26 mm. 

Evaluating the appropriate length of the screws is difficult because the sensation of the second cortex cannot be determined while drilling the K-wire into place, and x-ray evaluation is dangerous because pronosupination movements can bend the K-wires. Therefore, the Authors recommend measurements during pre-operative CT scans to determine the diameter of radial head, as the screw’s length should never exceed that value. The Authors inserted the first screw in the largest fragment until the sensation of a stable fixation was reached; the K-wire was then removed, the stability of the fragment was rechecked, and then x-rays were performed. If the fracture was reduced and stable, then other fragments (if present) were treated using the same procedure. 

The rehabilitation program began the day after surgery with gentle self-assisted active movements. A sling was positioned to protect the elbow from valgus stresses for three weeks, and manual activities were forbidden for two months. 

### 2.3. Follow-Up and Functional Evaluation

All patients underwent x-rays and clinical examination at 1, 3, 6, and 12 months after surgery, and at the final follow-up. Functional evaluation was performed using the Mayo Elbow Performance Index (MEPI) and the Broberg and Morrey Rating System (BMRS) [19,20,21].

### 2.4. Statistical Analysis

Statistical evaluation was performed using the IBM SSPS Statistics Software Version 23 (IBM, Armonk, NY, USA) for MAC. The independent *t*-test and the Chi-squared test were performed to compare both ARIF and ORIF subjects using the variables Surgery Duration, Functional Outcome Scores (MEPI and BMRS), and Complications rate. A *p* value < 0.05 was considered significative.

## 3. Results

A total of 32 subjects (15 females and 17 males) were selected for the study using radiological criteria (standard AP and lateral x-ray views and CT scan of the elbow with 2D and 3D reconstruction) to classify the radial head fractures, according to Mason.

Between June 2007 and October 2015, 13 patients were treated with ARIF and 19 patients with ORIF. 

Patients’ age ranged from 14 to 65 years (mean of 44.16 years) and the time between trauma and surgery ranged from 2 to 64 days (mean of 8.1 days). 

A total of 13 subjects were treated with ARIF. The patients were five males and eight females, with a mean time between the date of the trauma and the surgery date of 8 days (range 3–24 days). The other 19 patients were treated with ORIF (12 males and 7 females), by Kocher approach and with a mean time between trauma and surgery of 7.25 days (range of 2–64 days).

Average follow-up time of all patients was 10 years (range 7–15). 

Functional outcomes (as determined by performance scores) were obtained for 32 patients at the last outpatient consult (Table 3).

Independent *t*-test and the Chi-squared test were performed with the variables to be compared and a *p* value < 0.05 considered significative. The mean MEPI score for the ARIF group was 98.077 (SD ± 4.34), while ORIF group reported 91.57 (SD ± 11.67). This indicates significantly better results for the ARIF group (*p* = 0.036); mean BMRS score was 95.61 (SD ±5.99 for ARIF) and 92.06 (SD ± 8.66 for ORIF) (*p* = 0.181). 

Mean duration of the procedure was 71.60 min for ARIF (SD ±24.96 and range of 36–135 min) and 81.66 min for ORIF (SD ± 36.73 and range of 29–170 min) (*p* = 0.808). 

### Complications

No intra operative complications were observed. A total of 15 patients reported post-operative complications, 3 for ARIF group (23.1%) and 12 for ORIF (63.5%) (Table 4).

In the ARIF group, two patients reported elbow stiffness (15.4%). One underwent arthrolysis and screw removal; the other one had symptoms of ulnar nerve compression, with positive Tinel sign and distal dysesthesia. The third patient showed a heterotopic ossification at the lateral epicondyle margin without functional deficit (7.7%). One patient underwent a second surgery with open arthrolysis and screw removal.

In the ORIF group, the most frequent complication was persistent mild pain, followed by moderate inconstant pain, crepitation during ROM of radial head, pronosupination limitations, mild supination deficit, and weakness when complete flexion of the elbow was reached. From these patients, two underwent a second procedure: the first underwent screw removal and arthrolysis eight months after the osteosynthesis, and the second patient underwent arthroscopic arthrolysis after 15 months, with satisfactory range of motion improvement.

## 4. Discussion

Arthroscopic treatment of elbow pathology is increasing over time; treatments include reduction and fixation of radial head fractures that include associated injuries, such as coronoid fracture or collateral ligament avulsion [8,22,23]. This technique offers multiple advantages: a complete view of the articular surfaces of radial head and coronoid is possible [17], as well as removal of small intra-articular fragments and treatment of trochlear chondral damages or small coronoid fractures that would otherwise require extensive open medial access. There is much less tissue damage with ARIF compared to ORIF; however, ARIF still remains a technically demanding surgery that requires a long learning curve and high technical skills. Complication rate is reported with a huge variability depending on the surgeon’s abilities [24]. 

The first arthroscopic treatment of radial head fracture was described in 2004 [25] for a fracture of the neck of the proximal radius in a child. The fracture was manually reduced and fixed by percutaneous K-wires under a direct intra-articular visualization [26]. 

In 2006, Rolla et al. [16] published a series of cases that included 6 fractures of the radial head (II, III and IV types according to Mason) which were reduced and fixed by percutaneous screws. The performed technique was described, with an indication to work in the anterior compartment to reduce the fracture, and then change the view position to the posterolateral portal, creating an anterolateral portal for fragment fixation.

In 2007, Michels [15] described 14 cases, all classified as Mason type II. The authors performed ARIF using only two portals (anterolateral and posterolateral) and performed a small incision to insert the screws, with good clinical and radiographic results. Wang et al. [27] reported results from 18 cases with Mason type II fractures treated with percutaneous K-wires under arthroscopy with clinical good results. In 2019, Haasters et al. [28] reported a retrospective case series of 20 patients, highlighting good results and a high capacity of arthroscopy to diagnose and treat concomitant elbow injuries that might not be visible at MRI or CT scan. 

As evidenced by the literature, ARIF of radial head fracture is a procedure that is growing in popularity but is not yet widely performed. With the present study, the authors aim to report their 10 years outcome of the arthroscopic radial head osteosynthesis and describe different surgical technique using different portals, depending on the fracture pattern. 

In comparison to open osteosynthesis, screw length is more difficult to assess because performing an intraoperative fluoroscopy with the tools in place is complicated. Generally, the screws range from 14 to 18 mm long and, in case of indecision, the Authors recommend choosing a shorter screw and checking at the end of fixation to avoid loss of reduction or K-wire bending while obtaining a satisfactory radiographic view. Studying a pre-operatory CT scan helps to determine appropriate screw length and avoiding excessive length.

The instruments should be chosen carefully, because drill and screwdriver might have very short results, especially when fixation is performed at the anteromedial portal. If these tools are not available, the arthroscopic cannula can be cut and shortened as necessary before being inserted into the joint.

Another important aspect of ARIF in comparison to ORIF is the surgical time. In the present study, the mean surgical time for ARIF was 74.46 min, 2.64 min faster than ORIF group; however, the independent *t*-test showed no statistically significant difference (*p* = 0.808) between the groups, indicating that after the proper learning curve even a technically demanding procedure can be performed with confidence and reproducibility. Obviously, considering the difficulty of this procedure, the learning curve is steep and requires high elbow arthroscopic skills. 

The different position described in this paper, along with the findings in the literature that indicate how the radial head can be easily reached for screw insertion [29], aim to give elbow surgeons the tools they need to approach and improve this fixation technique. The Authors find it easier to perform the synthesis from the anteromedial portal when possible, with supination or pronation helping to reach a wider radial head surface. The AM working position gives the surgeon a wider articular space to work in without the need of an additional portal that may cause a tear (even if small) to the annular ligament. In this working position, the Authors suggest that it is more comfortable to also insert multiple screws in different directions, either parallel to the radial head surface or oblique through the radial neck. One disadvantage is the larger distance required to reach the radial head from the AM portal. Longer screwdrivers and drills may be necessary, especially in obese or highly muscular patients.

For the same reason, the Authors do not suggest the PL working position. As the capsule is very close to the radial head, it is more difficult to perform and the cartilage is easier to damage while reducing the facture. However, this position uses safer portals than anterior ones as they are far from neurovascular structures.

The AL working position can be considered a good balance between advantages and disadvantages. The space is narrow, and a small tear of the annular ligament occurs frequently, but the AL is in the same position that is used to reduce the fracture, and instrument length does not create difficulties.

In the present series, regarding the functional outcome, MEPI scores were better in the ARIF group (mean 98.07, *p* = 0.036) with statistical significance. In comparison to the previous series about arthroscopic reduction and internal fixation of radial head fractures, the presented results are comparable with results published by Rolla et al. [16], which reported three excellent and three good results for functional outcome. The present study reported 92.30% excellent and 7.69% good results (n = 13 patients, 12 excellent and 1 good). Another functional parameter used was the BMRS score, which had no statistically significative difference between the arthroscopic and the open groups (*p* = 0.181; mean score 95.615 and 92.063, respectively) and in comparison to Michel’s study (mean 97.6 and range 86–100 points; 11 excellent and 3 good results) [15], the present study reported similar values for the ARIF group (mean 95.61 and range 81–100 points; 10 excellent (76.9%) and 3 good (23.1%). 

Even if complications are reported in radial head ARIF, the present series included no neurological or vascular lesions other than the patient with symptoms of ulnar nerve compression. The major complication of this technique is the same as that described for elbow arthroscopy, i.e., the risk of nerve or brachial artery injury, but the frequency is not well known [5].

In the present series, the ARIF cohort had a complication rate of 23.1%. The rate of neurological complications is between 0 and 14%, according to El Hajj et al. [30], with more than half of the cases including the ulnar or median nerves. This complication can occur due to the proximity of the radial, posterior interosseous, ulnar, and median nerves to anterolateral and anteromedial portals [5]. These injuries can also be a result of laceration or be secondary to compression from a cannula, fluid extravasation, exposure to local anesthetics, or tourniquet-related problems [31]. The series by Kelly et al., a retrospective review of 473 consecutive elbow arthroscopies performed during an 18-year period, had an overall complication rate of 12%, with serious complications in less than 1% (permanent nerve lesions or infection), and minor complications (such transient nerve lesions) in 11% of the arthroscopic procedures [32]. This matched the presented complications of this study, with exception of heterotopic ossification, which was not analyzed.

The major limitation of this study is its retrospective design. However, to our knowledge, the existing studies have not reported results and complications of radial head ARIF with this long of a follow-up. As a retrospective study, the patient selection also represents a possible source of bias, in that the Authors tried to avoid enrolling all the patients that met the inclusion criteria.

## 5. Conclusions

The described radial head ARIF surgical technique represents a reproducible and safe procedure. A long learning curve is necessary, but with the proper experience it represents a tool that might be beneficial for patients, allowing a radial head fracture treatment with minimal tissue damage, evaluation and treatment of the concomitant lesions, and with no limitation of screws positioning. The surgeon must perform accurate pre-operative planning and intraoperatively choose the better working position so that they can fix all fracture patterns and obtain good clinical results, comparable to the state of the art radial head ORIF.

## Figures and Tables

**Figure 1 jcm-12-01558-f001:**
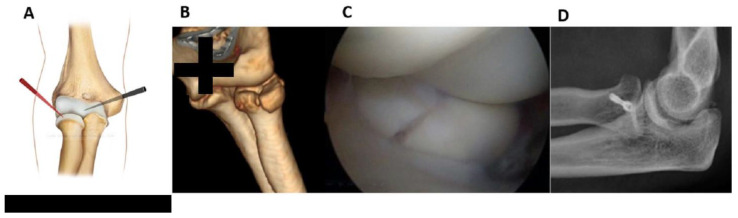
Fractures of the lateral half can be fixed from the anterolateral working position (**A**)**.** (**B**): pre-operative 3D CT scan showing the fracture. (**C**): intraoperative image of fracture fixation. (**D**): Control x-ray.

**Figure 2 jcm-12-01558-f002:**
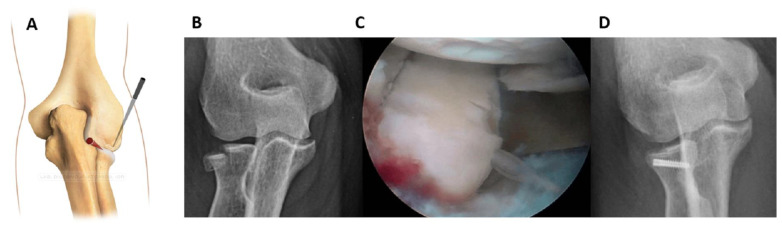
Fracture of lateral half of radial head that is fixed using the posterolateral working position; posterolateral and midlateral portals were created (**A**). From the superior view, the shaver and probe created the necessary space to reduce the fragment. (**B**): Pre-operative x-ray showing the fracture. (**C**): Intraoperative image of fracture fixation. (**D**): Control x-ray.

**Figure 3 jcm-12-01558-f003:**
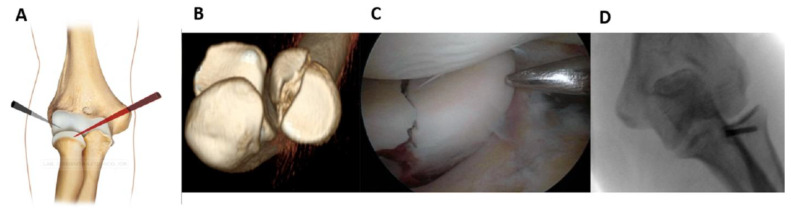
For the medial half fractures, the working position AM was more appropriate; the field was viewed from the anterolateral portal and the cannula was inserted through the anteromedial portal (**A**). (**B**): Pre-operative 3D CT scan showing the fracture. (**C**): Intraoperative image of fracture fixation. (**D**): Control x-ray.

**Table 1 jcm-12-01558-t001:** Patients’ demographics.

Demographic Data of the Study Population	ORIF	ARIF	TOTAL
**Patients (no.) (%)**	19 (59.41)	13 (40.62)	32
**Age (years)**	43.31 (±11.03/15–61)	45.38 (±16.37/14–65)	44.16 (±13.57) (14–65)
**Gender (M/F) (%)**	12/7 (63.25/36.84)	5/8 (38.54/61.52)	17/15 (53.12/46.92)
**Classification (Mason II/III) (no.) (%)**	12/7 (63.2/436.83)	11/2 (84.63/15.42)	23/9 (71.93/28.12)

Legend: no.—number; M—Male; F—Female; ORIF—Open Reduction Internal Fixation; ARIF—Arthroscopic Reduction Internal Fixation.

**Table 2 jcm-12-01558-t002:** Description of arthroscopic working position to address fixation of radial head fragments.

Working Position	Direction of Screws Insetion	Camera Portal Position	Probe/Elevator Portal Position	Portal of Screws Insertion
PL	Posterolateral	Posterolateral	Midlateral portal (soft spot)	Direct radial head accessory
AL	Anterolateral	Anteromedial	Anterolateral Ant lat proximal Midlateral	Direct radial head accessory
AM	Anteromedial	Anterolateral	Ant lat proximal Midlateral	Anteromedial

**Table 3 jcm-12-01558-t003:** Clinical and functional results of ARIF and ORIF.

	ARIF	ORIF
**MEPI Score**	98.07 (SD ± 4.34)	91.57 (SD ± 11.67)
**BMRS Score—Variables**	95.61(SD ± 5.99)	92.06(SD ± 8.66)
**Duration of the procedure (minutes)**	71.60 (SD ± 24.96) (range 36–135)	81.66 (SD ± 36.73) (range 29–170)
**Follow-up (years)** **(SD; range)**	10.14 (±4.32; 7–15)	10.52 (±26.46/7–15)

**Table 4 jcm-12-01558-t004:** Post operative complications and reoperations.

	ARIF	ORIF
**Complications**	3 (23.1%)	12 (63.5%)
**Stiffness and mild pain**	2 (15.4%)	4 (21.1%)
**Lateral ossification**	1 (7.7%)	0
**Inconstant pain**	0	1 (5.3%)
**Crepitation**	0	1 (5.3%)
**Moderate supination deficit**	0	1 (5.3%)
**Moderate pronation deficit**	0	1 (5.3%)
**Weakness in flexion**	0	1 (5.3%)
**Weakness in pronosupination**	0	1 (5.3%)
**Occasional lateral elbow pain**	0	1 (5.3%)
**Temporary cutaneous numbness**	0	1 (5.3%)
**Second surgical procedure**	1 open arthrolisis and screw removal	2 arthroscopic arthrolysis with one screw removal

## Data Availability

The data presented in this study are available on request from the corresponding author.

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
