# Peer review of "Elbow Arthroscopy for the Treatment of Radial Head Fractures: Surgical Technique and 10 Years of Follow Up Results Compared to Open Surgery"

_jcm, 2023, doi:10.3390/jcm12041558_

Round 1

Reviewer 1 Report

This is a very interesting retrospective study. But there are several questions:

1.      Problems with punctuation:

a)      Line 19, “Methods;” should be “Methods:”.

b)     Conflicting use of decimal points in the article, Line 21, 25, 26 and so on. In some countries decimal points can be replaced by commas, however this does not conflict with the consistency of punctuation use in scientific papers.

c)      Table 1 and 3. Either keep two decimal places or one, please be consistent.

2.      The specific statistical analysis method you used should also be in the results, along with the p-value.

3.      Another very important point is that the article does not mention selection bias and how to try to avoid it, although this is one of the characteristics of retrospective studies. The physical health of the patient at the time of selection of the procedure may have contributed to bias in the results.

4.      Can type II and III be discussed together and is there any statistical analysis to prove that the different distributions of the two types do not affect the conclusions?

5.      In fact, elbow arthroscopy for radial head fractures has been around for many years, so it is hoped that more critical thinking in the discussion.

Author Response

The Authors wish to thanks the Reviewer for its comments that helped to improve the quality of the manuscript. Attached you can find the response to the Reviewer's comments and the change made in the text accordingly.

Best Regards,

marco Cavallo

Reviewer 2 Report

There are spelling and sentence construction problems throughout this paper. They need to be corrected.

Table 1 - R and L hand information adds nothing and should be eliminated. Of no consequence or interest to include that info.

2.2.1 - Line 110 - What about those difficult reductions (those pushed through the capsule, into the PRUJ, or just not reducible? Was the joint not opened ever with the ARIF group? They are hard to reduce open let alone through a scope. Are the reductions difficult or VERY difficult? Please make clear as these are tough fractures to openly reduce?

3 Results - Line 186-189 - Remove info on handedness as it makes no difference.

4 Discussion - Line 222 - How do you know the reduction was more anatomic than open reductions where one can SEE the reduction and FEEL a reduction? Explain this statement as this reviewer feels it is false.

                     - Line 224 - spelling and other errors throughout.

Author Response

(The authors gave the same response as above.)

Round 2

Reviewer 2 Report

Good revision - no deficiencies noted

Author Response

The Authors wish to thanks the reviewers for the precious suggestions.

Best Regards

Marco Cavallo